# The Benefits of Dog-Assisted Therapy as Complementary Treatment in a Children’s Mental Health Day Hospital

**DOI:** 10.3390/ani12202841

**Published:** 2022-10-19

**Authors:** Elías Guillen Guzmán, Laia Sastre Rodríguez, Pilar Santamarina-Perez, Laura Hermida Barros, Marta García Giralt, Eva Domenec Elizalde, Fransesc Ristol Ubach, Miguel Romero Gonzalez, Yeray Pastor Yuste, Cristina Diaz Téllez, Soledad Romero Cela, Laura Real Gisbert, Miquel Salmeron Medina, Anna Ballesteros-Urpi, Astrid Morer Liñan

**Affiliations:** 1Department of Child and Adolescent Psychiatry and Psychology, Institute of Neurosciences, Hospital Clinic Barcelona, 08036 Barcelona, Spain; 2Programs and Projects in Animal Assisted Interventions, 08041 Barcelona, Spain; 3Centro de Investigación Biomédica en Red de Salud Mental (CIBERSAM), 08036 Barcelona, Spain

**Keywords:** animal-assisted therapies, dog-assisted therapy, mental health, children

## Abstract

**Simple Summary:**

Animal-assisted therapy is a rehabilitation and treatment procedure that uses trained animals to help people cope with difficult situations in addition to other therapeutic interventions. The most commonly used animals are dogs and horses. The current study analyzes the benefits of dog-assisted therapy in a pediatric population attending a mental health day hospital. This program was associated with positive impacts on factors including reduced emotional and behavioral outbursts and improved socialization behavior observed by therapists in the day hospital setting. Whereas the parents did not observe significant improvements outside the unit. Attendance rate and commitment to treatment also improved. According to the impressions of the health professionals, the dog-assisted therapy facilitated the motivation of the patients to therapy and improved the atmosphere of the unit. Dog-assisted therapy is a promising intervention that benefits children with mental health disorders, their families and mental health professionals. Its implementation is necessary as a complementary program of the treatment at a psychiatric day hospital.

**Abstract:**

Dog-assisted therapy (DAT) has shown benefits in people with mental health disorders. A child psychiatric day hospital would be a suitable setting to implement DAT and evaluate the benefits in a pediatric population. Methods: Mixed methods research in a naturalistic setting was considered in this pre-post quantitative study including 23 children under 13 treated in a day hospital over 2 years. Quantitative analysis included the number of emotional and behavioral outbursts and attendance rate and self-control and social impairment questionnaires completed by family members and therapists. In the qualitative study, the experiences of 12 mental health professionals involved in DAT were documented through semi-structured interviews. Results: On DAT days, there were fewer emotional and behavioral outbursts and higher attendance. Significant differences were obtained between pre- and post-test scores on the SCRS and the SRS-2 completed by the therapists, while no significant differences were obtained on the questionnaires completed by the parents. Observations based on the qualitative study were as follows: (1) DAT improves emotional self-regulation; (2) DAT could facilitate the work of therapists in day hospitals; (3) health professionals displayed uncertainty due to a lack of familiarity with DAT. Conclusions: DAT improved emotional self-regulation, attendance rate and self-control and social response in children with mental disorders attending a day hospital.

## 1. Introduction

Mental health problems presented by children and adolescents are treated at primary care and child/adolescent mental health centers. Occasionally, in cases where the patient is suffering from a severe mental disorder (SMD), a more intense treatment in outpatient partial hospitalization services, called a day hospital (DH), may be necessary [1]. 

The treatment at a DH is intensive and rehabilitative and it is based on an individualized treatment plan that takes into consideration the specific needs of the patient [2]. It is conducted by a team using an interdisciplinary and comprehensive approach. It is less restrictive in terms of admission than an inpatient psychiatric ward. The treatment, however, is more intensive than that in an outpatient unit. The objective is to provide the patient with a structure and an intensification of the therapeutic intervention that facilitates improving their psychopathological stability while favoring their integration into the community in a safe way. 

Patients attending a DH present a wide variety of psychopathological disorders: neurodevelopmental disorders (autism syndrome disorder (ASD), attention deficit hyperactivity disorder (ADHD), etc), attachment disorders, behavioral disorders, affective disorders, self-harm and suicidal behavior, incipient psychosis and symptomatology that can evolve into a personality disorder. In such disorders, problems may include: low self-esteem, problems in establishing social bonds with others (isolation, over possessive relationships, overzealousness in relationships, dominance in relationships or excessively submissive attitudes), difficulty in expressing emotions and regulating them (difficulty in identifying feelings, difficulty in expressing them correctly, and great difficulty in controlling anger and frustration), difficulty in following instructions, difficulty in generating self-instructions to redirect their own behaviors and lack of motivation. 

Animal-assisted therapy (AAT) is a rehabilitation and treatment procedure that uses trained animals to help patients cope with certain situations in a complementary way to other therapeutic interventions [3,4]. The most commonly used animals are dogs and horses [5,6]. 

Previous studies of ATT in young mental health populations [7,8], have investigated the positive influence on psychological, physiological and behavioral well-being [9,10,11]. Specifically, AAT has been used successfully to treat people suffering from mental disorders such as post-traumatic stress disorder (PTSD) [12], ASD [13,14], ADHD [15], depression and anxiety [16,17]. There is evidence confirming that AAT is associated with a reduction in stress and an increase in positive emotions [18]. Likewise, it has been shown to be a self-esteem enhancer, a catalyst for mood improvement [19] and a facilitator of social interactions [20]. In addition, AAT, specifically dog-assisted therapy (DAT), has been found to benefit the well being of healthcare professionals by improving the hospital atmosphere [21] and reducing perceived stress by staff [22]. 

One study showed similar findings in an inpatient pediatric mental health service [23], but no studies to date have been performed in an intensive community mental health program such as a Day Hospital for children and adolescents. For this reason, further research is needed to investigate the feasibility and the benefits of the incorporation of the DAT within the DH environment.

The present study was designed to assess the feasibility and the benefit of DAT on children suffering from mental disorders who are attending DH. The following hypotheses were raised: (1) DAT will improve the self-control and social skills of children suffering from a mental disorder during DH therapeutic activities. (2) DAT will facilitate patients in achieving their goals within the DH framework. 

## 2. Methods

Mixed methodology with a naturalistic setting [24], combining quantitative and qualitative research, was carried out in order to strengthen the conclusions of the study. 

The quantitative study included children under 13 years old who were assessed before and after DAT treatment at a DH, with a comparison of attendance rate and emotional outburst between DAT versus and non-DAT therapy days. In the qualitative study, a hermeneutic phenomenology [25] was applied, describing the experiences of health professionals with the incorporation of DAT and its impact on work at the DH.

### 2.1. DAT Intervention

Children under the age of 13 come to DH on Tuesday and Thursday for individualized treatment. DAT is included in the DH activities only on Thursdays. There was no DAT offered on Tuesdays. The DAT was used by professionals of service to carry out group therapy and weekly individual visits by the therapist to the users who were cited for DH. 

The program comprises activities that help to achieve therapeutic objectives which are categorized into two large blocks:Creation of the bond (professional, patient and dogs) consisting of three areas:Verbal and non-verbal communication activities. Example: Sharing questions in relation to the animal and other topics of interest using treats.Sensory activities that are performed through touching (petting and brushing the dog).Low complexity motor activities to observe and work on patient-dog interaction. In these activities, the child asks the dog to perform skills (shake its paw, high-five, make the dog roll over, etc.) and reward the animal, which helps gain confidence and security.Personal development: activities related to taking care of the dogs (hygiene and feeding), their training, and physical and social-cognitive skills (playing, walking and emotional communication). All interventions involving activities with the dogs were adapted to the specific needs of each patient. Some activities carried out during the study were as follow:Food activity: the activity consisted of classifying pictures of food according to whether they were appropriate or not for feeding the dogs. After that, each patient completed the same classification for themselves. Once this classification was completed, the patient described the characteristics of the food (texture, color, etc.…) that they like or dislike. This is an activity more frequently used for children with ASD.Social cognitive activity: in turns, each patient prepared a circuit using balls, hoops, mats, or cones that the dogs had to perform under supervision of the professionals. Their companion had to pay attention to and memorize the circuit in order to describe and imitate it using another dog.Emotional relaxation and bonding activity: with the dogs lying/resting next to the patients, they petted the dogs, practiced breathing in sync with the dogs, and listened to relaxing music in order to create a mindful atmosphere and a calm state of mind.

This activity was performed by a specialist trained in dog assisted therapy (ED) and one of the professional staff from the DH (the staff member had K-Partners recognition and was certified in the CDAT method of applying the intervention). In all sessions there was a member of the research team supervising the interventions.

The wellbeing of the dogs was a priority during the time that the DAT was administered (see Appendix A for more information). All sanitary and hygienic protocols of the Center for Dog-Assisted Therapy, CDAT (Centro de Terapia Asistida por Canes-CTAT) were followed. The CDAT is a private health care center that specializes in the field of dog training and assisted therapies, with previous experience with the same interventions in other hospitals in Barcelona.

The entity (CDAT) complies with and applies the IAHAIO white paper document (reference entity in the application of animal in relation to interventions that unse the human animal bond). Likewise, the professional comply with the animal welfare standards dictated by IAHAIO.

In the intervention five dogs participated in the activities: three medium-sized (two Golden retrivers and one Labrador retriever, weighing approximately 30 kg) and two small (Cavalier King Charles spaniels, weighing approximately 12 kg). They were between two and five years old. All dogs passed a K-Partners for CTAC, physical fitness and therapy dog qualification.

### 2.2. Phase 1: Quantitative Study 

#### 2.2.1. Sample

The study was proposed to 25 patients who attended the DH during January 2020 and December 2021. All agreed to participate. During the study there were two dropouts (one by consensus of the therapist and another for not completing the questionnaires). The final sample consisted of 23 participants.

The inclusion criteria were patients under 13 years of age that attending children´s DH. Patient with low-functioning ASD were excluded.

All subjects were invited to participate in the study. Families were explicitly informed about the voluntary nature of the study, their rights the procedures, the possible risks and benefits involved, and the voluntary nature of the study. A WA written consent form was required from parents or legal guardians; the children gave their consent in writing after the nature of the procedure was fully explained to them.

Written informed consent was obtained from the participants before starting the study, which was approved on March 4, 2019, by the Ethics Committee of the Hospital Clinic of Barcelona in accordance with the Declaration of Helsinki of 1975 revised in 2000 (reference numberHCB/2018/0652). 

Sociodemographic and clinical variables of the sample were registered (see Table 1). 

#### 2.2.2. Assessment

The emotional and behavioral outbursts of the patients were quantified during the 2 years of the study (except for a 4 month period during the covid lockdown). In addition, the DH attendance rate was recorded. The DAT was carried out once a week, such that a comparison could be made of outbursts and attendance between the DAT days and non-DAT days. 

Families and therapists completed two pre- and two post-treatment questionnaires about the patients as part of the evaluation process.

The Self-Control Rating Scale (SCRS): This test includes 33 items designed to measure the extent to which a child’s behavior can be described as self-controlled versus impulsive. Children with greater self-control difficulties are presumed to have a higher score. It is stated that patients who obtain a score of 160 to 165 or higher are candidates for treatment to improve self-control. The SCRS has been studied with several samples of children of primary school age and has shown excellent internal consistency (α = 0.98) and good stability according to the test retest correlation (r = 0.84, *p* < 0.05) [26]. In this study, the experimental version in Spanish by Garcia i Giral M, Berltán S, and Nicolau R. (1990) was used [27]. 

The Social Responsiveness Scale-2 (SRS-2): This test is a standardized and referenced caregiver report questionnaire for documenting patterns of ASD symptomatology. It has a high sensitivity for detecting milder degrees of social impairment. For the assessment, caregivers are asked to rate 65 items about the subject’s behavior using a Likert scale from 1 (not true) to 4 (almost always true). The SRS-2 items are grouped into five factors: social awareness, social cognition, social communication, social motivation and restricted interests and repetitive behaviors. This scale has shown cross-cultural validity [28,29]. The SRS-2 form for school-age children has strong psychometric properties in clinical and nonclinical standardization samples. The internal consistency and reliability are α = 0.95 and 0.97, respectively. The SRS-2 items were adapted to the Spanish language following the guidelines proposed by Muñiz [30]. 

#### 2.2.3. Statistical Analysis

Data were analyzed using SPSS. Descriptive statistics were used to describe the sociodemographic characteristics of the participants and the research variables. For the questionnaires, the mean and range of scores were calculated. Differences in scores before and after treatment and between attendance and emotional outburst between DAT days vs, none were analyzed using paired sample *t*-test. 

### 2.3. Phase 2: Qualitative Study 

#### 2.3.1. Sample of Health Professionals

In the qualitative study, a convenience sampling was performed, in which a sample of professionals working at the DH was selected (*n* = 12; see Table 2). These health care professionals were involved in the DAT and signed an informed consent to participate. The health care professionals were clinical psychologists, psychiatrists, mental health nurse specialists, social workers, and psychology and nursing residents (Table 2). Interviews were conducted by the research team. 

#### 2.3.2. Assessment 

The principal research instrument was a semi-structured interview (in Spanish) following the structure of previous publications [31,32]. The interviewer encouraged the health care professionals to talk about their experiences with DAT. All interviews were conducted face-to-face and audio recorded. These interviews were transcribed and subsequently verified by the professionals interviewed. The participating health care professionals signed the informed consent to record the interviews. Each interview lasted approximately 30 min. Data collection and analysis continued until theoretical saturation was reached, i.e., the additional interviews did not provide new insights for the analysis. Information was also obtained from notes made by the principal investigator. 

#### 2.3.3. Analysis

The analysis of this study was carried out using these four steps: (1) open coding where the main researcher read each transcript of each interview line by line, recording notes to determine the categories; (2) the principal researcher together with another researcher reviewed and discussed the main topics; (3) axial coding, where all interviews are read a second time and both researchers detected the associations between themes and sub-themes related to the context and content. They related all transcripts of the completed interviews to strengthen meanings and arrive at a theoretical construct; (4) integration, where the central themes or main categories that emerge from the data were conceptually reordered and put back into context [33].

## 3. Results

### 3.1. Results of the Quantitative Study

Twenty-three boys and girls (mean age 10.33, range 7–13 years) participated in the quantitative study. 48% (11 out of 23) had ASD and 17% had a conduct disorder.

Results showed statistically significant differences in attendance rates between the days in which DAT was carried out vs. not. Specifically, attendance on DAT-days (M = 98.04; SD = 2.21) was higher than attendance on days that DAT was not perfomed (M = 92.3; SD = 10.78) (t(20) = 2.6, *p* = 0.016.

Regarding emotional outbursts, there were statistically significant differences on the days that the DAT was performed vs. those that the DAT was not performed, where the scores on the latter (M = 1.71; SD = 2.49) were greater than those on the former (M = 0.48; SD = 0.68) t(20) = −2.77, *p* = 0.012).

As shown in Figure 1 and Figure 2, there were statistically significant differences in the mean scores before and after treatment in the SCRS [132.5 vs. 110.83, t(22) = −2.88, *p* = 0.009)] and in the SRS [t(22) = 2.81; *p* = 0.010] filled out by the therapists. The questionnaires completed by the parents or legal guardians did not show significant differences before and after treatment. Figure 1 and Figure 2 present the data. 

### 3.2. Results of the Qualitative Phase 

The qualitative study allowed us to make three main observations: (1) DAT improved emotional self-regulation; (2) DAT could facilitate the therapists’ work at the DH; (3) the health professionals displayed a sense of uncertainty due to their lack of familiarity with this type of therapy. 

**Observation** **1.***DAT improved emotional self-regulation in children undergoing DH treatment: “Patients activate fewer defenses in the presence of dogs”*. 

Nine of the 12 health professionals stated that DAT helped to improve some aspects of emotional regulation and management. 

“I have worked mostly with groups, and with DAT I observed that the patients’ defenses and resistance were lowered, therefore, it was easier to work with them. At a therapeutic level, they are calmer and trust the therapists more. Another aspect that I think is also important to highlight is the work climate that day was calmer, not only for patients but also for DH professionals. I think that also influenced the patients themselves”.

Research participants described situations with patients in which the intervention with animals helped emotional self-regulation. 

“I also remember a girl diagnosed with attachment disorder who was experiencing psychomotor agitation at the time her mother was leaving. The dog appeared and calmed her down. It was restorative for her”.

**Observation** **2.***DAT could facilitate the therapists’ work at the DH: “the therapy mad individual and group sessions with patients easier”*. 

The 12 participants of the qualitative study described how DAT aided them in their work at the DH. 

“DAT has helped us work through and improve the technique of exposure with response prevention. The dogs have made it easier for the patient to approach situations that generated a lot of anxiety”. 

“In therapy, dogs act as facilitators they are simply there and do not judge. I believe that this helps the patients feel more comfortable. They improve at a behavioral level and at an emotional level. They are more relaxed”. 

An important element that the research participants mentioned in DAT meetings is that it helped improve the atmosphere at the DH unit. This promoted a warm environment. 

“DAT creates a more relaxed atmosphere. This allows us to delve into certain aspects that cannot be addressed on a more normative visit”.

Some of the research participants also said that DAT helped improve the patient’s bond with the DH team. 

“The atmosphere that was generated during the sessions was very pleasant and I think it can be useful to improve treatment and the therapeutic relationship.” 

“I remember working with a patient. The therapeutic relationship with her was not going well, and she was not having a good time, so I suggested that she come in for DAT. We did three sessions, the patient and her mother participated, and it went very well, I think she was more motivated to come to the hospital and the therapeutic relationship also improved”. 

Four research participants observed a positive change in the social interaction between the patient and the professionals. 

“The whole issue of social interaction has improved a lot, especially in my unit, which works with children with autism and is one of the most affected psychopathological areas. The main issues are social interaction and communication. We have come across children who had no way of expressing themselves and yet through DAT and treating the dog as if it were this third person, we have managed to detect situations of bullying they had suffered”.

**Observation** **3.***The health professionals had a sense of uncertainty due to their lack of familiarity with this type of therapy: “I had not been able to work with this type of therapy before and I did not know exactly how it worked”*. 

Most of the research participants did know about DAT, through research studies, social networks or conversations with other professionals, but many of those interviewed had never worked with DAT. 

“I had not been able to work with this type of therapy before, but if I had heard of it, it was performed in other units. But I also did not know exactly how it functioned and how it could influence the therapeutic process”. 

Three professionals reported that they knew and had experience with DAT. They had worked and/or completed trainings with DAT in care settings other than in the DH. 

“Yes, I was aware of DAT, because in another place where I worked in the past, therapy with dogs was also carried out, but not in the same area as we are now. Before, I was in the field of children with multiple disabilities”. 

## 4. Discussion

This study was designed to assess the feasibility and benefits that DAT may have on the recovery of children with mental health disorders who attend a DH. To our knowledge, this was the first study to investigate the effects of DAT in children with mental health disorders within an intensive community program such as a DH. The DAT was carried out for 2 years and was managed by mental health professionals (psychologists, psychiatrists, social workers and nurses specialized in mental health care) and the trainer in dog-assisted interventions.

The results show that the children who participated in the study and received DAT interventions presented a lower number of emotional and behavioral outbursts during their DH stay, compared to the days that DAT interventions were not carried out. These results may be indicators of a better self-control capacity of the boys and girls. In addition, their attendance rate increased on the days when DAT was performed, improving and facilitating adherence to therapeutic treatment. Regarding the results of the questionnaires, significant differences were observed in the scores of the therapists, in contrast to no differences in the parents’ reports

The subjective perceptions of the professionals regarding the efficacy of DAT supported the findings of the quantitative study. The professionals affirmed that DAT favors bonding, improves the atmosphere, increases motivation, improves emotional self-regulation and helps improve communication and thus social interaction.

Parents results about their children’s behaviors were not significant. This may be due to the observed changes being more evident in the Day Hospital setting and were not generalized outside the scope of DH. However, much remains to be understood about the non-significant findings according to the parents.

Current hypotheses about the therapeutic benefits of DAT are consistent with the idea that human interaction with dogs favors the success of therapy [34,35]. Moreover, developing a bond with an animal helps the participants gain new skills that can lead to positive changes in cognition and behavior [36]. The benefits and improvement could be due to the calming effect and reduction in stress associated with the presence of the animals [37].

Currently, there are studies confirming that dog programs implemented in hospitals are promising complementary interventions that benefit staff, patients and their families [38,39].

Merriam-Webster defines “best practice” as “a procedure that research and experience have shown to produce optimal results and that is established or proposed as an appropriate standard for widespread adoption” [40]. Therefore, carrying out an effective DAT intervention as an aspect of health care must be based on scientific evidence and demonstrate quality results. DAT must continue to grow and be studied over time in order to demonstrate its effectiveness as a part of community mental health services.

## 5. Limitations

This study was carried out in a natural setting in the context of daily clinical care, which means that neither the therapeutic treatment nor the activity of the unit was modified. The implementation of DAT did not imply the cessation of conventional therapies. This precludes the possibility of demonstrating that the DAT intervention was the sole cause for the positive results on the self-control and social skills questionnaires. 

It is important to note that the scale used was SRS-2 which is usually used for the diagnosis of ASD, which could have negatively influenced the efficacy of the quantitative results of the study. The qualitative study was carried out in order to compensate for some of the limitations involved in the quantitative study. This is a valid point for the group of therapists. However, for the parents, only quantitative data were available. Much remains to be understood about the non-significant findings of this group.

It should be noted that the results obtained in this study are not generalizable outside the HD setting.

Another unexpected limitation was the appearance of COVID-19. This setback resulted in a 4 month delay in 2020 and the necessity to comply with new restrictions, such as limiting the maximum number of participants in the group and the mandatory use of face masks, which may have been detrimental to human-dog interaction [41].

We view this as a pilot study with the complete understanding that it needs to be reworked and expanded in order to devise a study with more rigorous methodology in the future.

## 6. Conclusions

This mixed methodology study aimed to assess the impact of DAT in a DH and to describe the experiences of health professionals who incorporate dogs in their work process.

The children who participated in the study and received DAT interventions presented a lower number of emotional and behavioral outbursts during their stay in the DH unit. In addition, it was quantified that the days on which DAT was administered in the unit, the attendance rate of the children improved. The clinicians observed some improvement in self-control and social skills following the treatment, but this cannot be attributed to the incorporation of DAT into DH. With the results we obtained, it can be stated that the incorporation of DAT in DH is feasible. It is necessary to continue developing and expanding studies with a more rigorous methodology. 

## Figures and Tables

**Figure 1 animals-12-02841-f001:**
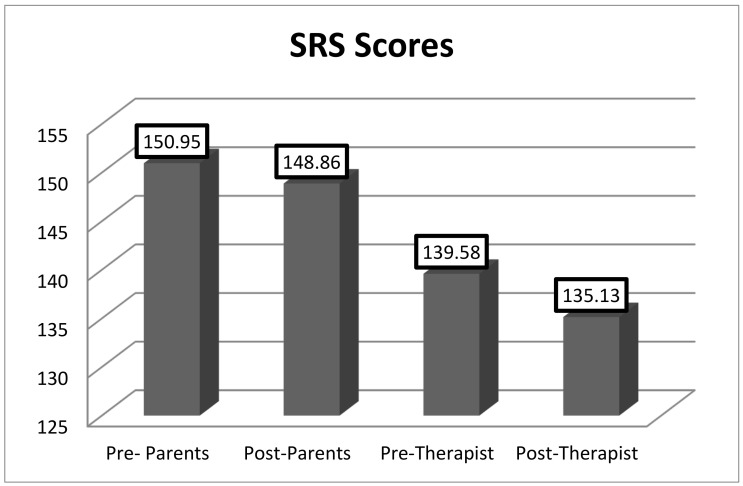
Raw scores of SRS completed by parents and therapists before and after treatment. Statistically significant (*p* < 0.05).

**Figure 2 animals-12-02841-f002:**
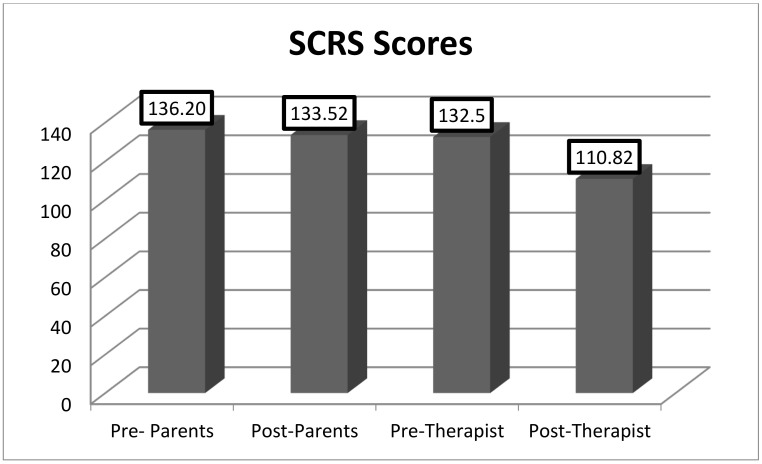
Raw scores of SCRS completed by parents and therapists before and after treatment. Statistically significant (*p* < 0.05).

**Table 1 animals-12-02841-t001:** Quantitative study: sociodemographic and clinical variables of the sample (*n* = 23).

Variables	Descriptive
Gender (%)	
Boys	19 (82.6%)
Girls	4 (17.4%)
Mean age in years, (SD) range *	10.3 (1.8), 7–13
Diagnostics (%)	
ASD	11 (48%)
Conduct Disorder	4 (17%)
Anxiety Disorders	2 (9%)

* Mean (SD), range. Other values are presented as *n* (%).

**Table 2 animals-12-02841-t002:** Qualitative study: sociodemographic variables of the professionals who participated in the study (*n* = 12).

Variables	*n* (%)
Gender	
Male	1 (8.34%)
Female	11 (91.66%)
Occupation	
Clinical psychologists	5 (45.45%)
Psychiatrists	1 (9.09%)
Residents in training (psychologists, nurses)	3 (27.27%)
Social workers	3 (27.27%)

## Data Availability

The data presented in this study are available on request from the corresponding author.

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
