# Peer review of "The Benefits of Dog-Assisted Therapy as Complementary Treatment in a Children’s Mental Health Day Hospital"

_animals, 2022, doi:10.3390/ani12202841_

Round 1

Reviewer 1 Report

This is an interesting paper that seeks to extend existing research on AAT by applying it in a (somewhat) new setting. Unfortunately, methodological limitations and incompleteness in both the study design and analysis/writeup hamper the paper’s potential for impact.

The paper contains a number of unfounded or shakily-founded claims, beginning with, “In most cases, this treatment alone is enough to bring about a significant clinical improvement” (no citation given, and a very strong statement about the entirety of outpatient mental health treatment for children and adolescents).

Lines 71-81 fail to present an adequate or accurate picture of the mixed body of research findings on the efficacy and effectiveness of AAT.

Lines 82-85 are missing citations for the studies mentioned there – even if there is not much background research in DH settings, what does exist should be described much more thoroughly.

The methodology of the AAT intervention is described very loosely, and the researchers acknowledge that there was no oversight over how the intervention was conducted. Even allowing for flexibility in customizing interventions to patients, this does not inspire confidence that the delivery of the AAT was really “a thing,” and certainly not one that could be replicated with fidelity should anyone attempt to extend this pilot research. At the very least, information should be provided about how frequently (on average) each component of the intervention was delivered.

The quantitative analyses are insufficient. The only inferential statistic provided is the paired-sample t-test pre-post treatment, which could easily reflect general improvement in functioning as a result of treatment overall and/or regression to the mean rather than any effect of the AAT intervention. Further, “the questionnaires completed by the parents or legal guardians did not show significant differences between pre- and post-treatment,” which is omitted from the abstract and discussion and should not be glossed over—the researchers have conflicting findings, and need to address them instead of cherry-picking the ones that fit the hypothesis. No statistics beyond descriptives are provided to compare the only two findings that are potentially noteworthy: attendance and number of outbursts on AAT days. These should be included, and the discussion should address that the study may have been statistically underpowered.

The lack of a comparison group is the most significant limitation to the study, and my reason for a Reject recommendation. The only place this is addressed is in a single line of the Limitations: “A clinical trial was not possible due to time, cost and ethical restraints.” For such a major methodological limitation, more needs to be justified here. If a full clinical trial was unethical due to depriving some participants of treatment, could a wait-list control (at least for the AAT component) not have been conducted? In fact, the Limitations section states that COVID forced a cap on the number of participants in the group; sounds like there was a ready-made control group consisting of those who couldn’t participate. Or could patients of a comparison DH with no AAT program not have been matched? I am aware of at least one randomized controlled trial of AAT that included patients in a day hospital setting, albeit with a small sample size (Stefanini et al., 2015). How does this study significantly extend that previous research?

Should the authors choose to submit this work to another journal, I would recommend focusing on the qualitative analysis and minimizing or omitting the quantitative analysis. If included, it should be with much greater attention to its limitations.

Author Response

Dear reviewer 

First of all, I would like to thank you for the time you have invested in reviewing this article and the annotations I received from you. They have helped me reflect. 

We have reviewed the methodology and we agree with the limitations you mention. Yes, it is true that it is a service where this type of intervention is carried out for the first time and we wanted to check if it was feasible to incorporate the DAT in DH. I would love to let you know that my team and I are already thinking about new projects in the future and your comments have helped us a lot in building the new studios. We are thinking of incorporating a control group. Hopefully it can be done without any impediment. 

We also agree with some revisions you make to the text and have tried to improve them. We agree that the document contains a number of unsubstantiated or unstable claims and we have tried to change them and improve the references. 

Regarding the statistical analysis, we have worked to improve it and thus obtain more objective data. 

Thank you very much for the contributions. 

Best regards 

Reviewer 2 Report

Overall comments

This manuscript was well-structured and -written. The use of mixed-methods design was praised for its ability to overcome the inherent limitations of each phase.

The reviewer suggested an alternative statistical test (split plot ANOVA) to better manage type I error when computing multiple simple t-tests. A Chi-square test is also recommended to add higher confidence to the conclusion that attendance increased in sessions with DAT.

The reviewer also suggested having more appropriate citations for the methods section. It is appropriate to refer to how past research was conducted. However, it is necessary to refer to the specific textbooks or guidelines for data analysis where their peer-review process focused on methodology specifically.

Specific comments

L27: It’s recommended that the authors use the term “Mixed-methods research” or its other formal variations.

L18-24, L33-38: Consider also mention the results from the carers.

L69. References 3,4. These are not appropriate sources to define types of AAT. They may cite a source that defines AAT. Please consider Howell et al. (2022) DOI: 10.3390/ani12151975.

L92. Use the phrase “mixed-methods design” consistently.

L94, L138-151, L221-222: Participants in the quantitative phase should be the parents and the therapists because the data is their observations and perceptions about the children. Participants are those who provide data. Apart from Table 1, data were from and about the therapist and parents (see the descriptions of the scales, to be completed by the adults and professionals).

L96. Inappropriate citation for data analysis. See comments on L218 below.

L99-135: Please suggest if licensing was required of the DAT provider, their qualification, experience and if the model was an established procedure with guidelines or if it’s an in-house program from the provider.

L187-188: The repeated t-test was used. However, given there are two groups of respondents (carers and professionals) and two-time points (pre and post), the reviewer wonders why a split plot ANOVA was not considered, which could provide further useful results about differences between carers and professionals and if there would be any interactions with the pre-post results.

L200-208. Please indicate the language used in this phrase, i.e., Spanish similar to the 1st phase, or English.

L206-208: Please provide a citation to the methods used here. The authors may consider textbooks from social sciences and psychology for more appropriate citations in qualitative research methods.

L218. References 29 and 30 don’t seem to be a guide for doing thematic analysis. Please check in those references which source they cited for their data analysis, or else please check the original source for thematic analysis and cite it here.

L228: Please comment on testing of the assumptions for the t-test if they were met.

L231-232: Please provide statistics results for all even the non-significant results. They will help inform future research. Also please consider comments for L187-188 above. If there are statistical reasons for not doing a mixed design ANOVA.

L314: This was based on descriptive statistics. It would be more appropriate to consider a Chi-Square test and base this conclusion on some inferential statistics.

L321: somewhere here please also discuss the non-significant findings of carers’ reporting of their child’s behaviours. Aspects to consider include a tentative explanation as to why it was the case, and if any possible confounding factors would play a role here.

L347: This is a valid point for the therapist group. However, for the carer group, only quantitative data was available. There is still much to understand about the non-significant findings of this group. Please comment on this in the Limitation section.

Author Response

Dear reviewer 

First of all, I would like to thank you for the time you have invested in reviewing this article and the annotations I received from you. They have helped me reflect and improve the manuscript. 

In the statistical analysis we have worked to improve it and thus obtain more objective data. 

I sincerely thank you for your input regarding citations and also at the level of methodological language. 

Thank you very much for the contributions. 

Best regards 

Reviewer 3 Report

General Comments

This is a well-written study on an important topic. The population and setting is unique and thus the paper could make an important contribution to the field. There are some concerns that must be addressed, and if they are addressed, I support publication of the paper. Here are my suggestions/concerns:

Animal Care and Ethics - I see that the study was approved by an institutional review board (IRB), which typically addresses ethics related to human well-being. Studies involving animals should also be reviewed by a body addressing ethics of and potential harm to animals – an institutional animal care and use committee (IACUC) or similar. Did that happen for this study?

A clearer description of informed consent/assent is needed. For minors, it is important to attain parental informed consent (which is included in the paper currently) and written or verbal assent from legal minors (which is not explained in the paper currently). Lines 145-146 should be expanded to clarify the procedure.

Recruitment - It is not clear how participants were recruited for the study – were all patients meeting criteria approached, and if so how? Of those approached, how many said yes and how many said no?

Introduction – The introduction is well-written and organized. It gives an effective introduction to day hospital treatment and AAT. It is lacking reference to studies or theories to justify the hypotheses. Hypothesis 1 predicts that DAT will improve self-control and social skills, but these specific concepts are not addressed in the introduction. There is no justification for why one would predict these specific outcomes.

Method Section – DAT Intervention (2.1) – Excellent description of the intervention. Many articles on AAT interventions lack details about human-animal interaction – what happens between the people and animals during the intervention. Your description clearly describes activities/behaviors that occurred and their purpose in the intervention.

Need more details on the dogs involved. Number of dogs, breed, size, weight, age, training, etc. Were the dog handlers trained/certified in a particular model of AAT? Were the dogs certified/trained?

Results – Lines 224-228 – Authors report differences in outbursts and attendance but do not site statistical analysis confirming the differences were statistically significant.

Discussion – Lines 312-314 – what is the comparison here? Children presented lower outbursts…than who or when?

Self-Control was named in the hypotheses but that term is not used in the discussion. Be sure to use consistent language throughout the paper in – intro, method, results, discussion – to make clear the connection from the literature, to the procedure, to the results, and to the implications.

Limitations – Line 343 – the study is referred to as quasi-experimental for the first time. Please state this earlier, at the end of the Intro, before the Method section. More specifically, I would call this a One-Group Pretest-Posttest Quasi-Experiment. Line 347-348 – the authors state the qualitative study was carried out to account for limitations of the quantitative study. That is not a very strong justification. A stronger argument and connection needs to be made between the quantitative limitations and qualitative methods/results. Line 353-355 – too vague. Be more specific on what is needed.

Conclusion – Line 358 – “dogs as a tool” is poor language choice. More animal affirming language should be used. Dogs were incorporated into an intervention. Practitioners partnered with Dogs. Using animals as a tool brings up questions about animal welfare and ethics.

Author Response

Dear Reviewer 

First of all, I would like to thank you for the time you have spent reviewing this article and the comments I received from you. They have helped me reflect. 

Regarding the Ethics and care of animals, the IAHAIO Whitepaper document has been applied and both the study and the CTAC comply with the animal welfare standards dictated by IAHAIO. 

Thank you for the notes on recruitment and informed consent, the intervention, and the conclusions. They are already entered in the manuscript. 

We have reviewed the methodology and agree with the limitations you mention. Yes, it is true that it is a service where this type of intervention is carried out for the first time and we wanted to check if it was feasible to incorporate the DAT in DH. 

Regarding the statistical analysis, we have worked to improve it and thus obtain more objective data. 

Thank you very much for your contributions. 

Best regards 

Round 2

Reviewer 1 Report

I appreciate the authors' effort in making revisions to the paper in response to the reviews received; these amendments and additions are thorough and clear. The paper needs one more round of editing for spelling and style, but is otherwise acceptable for publication in my view.